# The MADS-Box Transcription Factor EjAGL18 Negatively Regulates Malic Acid Content in Loquat by Repressing *EjtDT1*

**DOI:** 10.3390/ijms26020530

**Published:** 2025-01-10

**Authors:** Zhuoheng Chi, Luwei Wang, Qiankun Hu, Guangquan Yi, Shuming Wang, Qigao Guo, Danlong Jing, Guolu Liang, Yan Xia

**Affiliations:** 1Key Laboratory of Agricultural Biosafety and Green Production of Upper Yangtze River (Ministry of Education), College of Horticulture and Landscape Architecture, Southwest University, Beibei, Chongqing 400715, China; chizhuoheng0331@foxmail.com (Z.C.); 15839552412@163.com (L.W.); huqiankun0620@foxmail.com (Q.H.); masteryi2020@163.com (G.Y.); wangsm2018@swu.edu.cn (S.W.); qgguo@126.com (Q.G.); jingdanlong@swu.edu.cn (D.J.); 2Academy of Agricultural Sciences, Southwest University, Chongqing 400715, China

**Keywords:** loquat, MADS-box transcription factor, malic acid, fruit

## Abstract

Malic acid is the major organic acid in loquat fruit, contributing to the sourness of fruit and affecting fruit flavor. However, the transcriptional regulation of malic acid in loquat is not well understood. Here, we discovered a MADS-box transcription factor (TF), EjAGL18, that regulated malic acid accumulation in loquat. EjAGL18 is a nucleus-localized TF without transcriptional activity. The expression of *EjAGL18* increased during fruit ripening, opposite to the accumulation pattern of malic acid in loquat. The transient overexpression of *EjAGL18* in loquat fruit downregulated malic acid accumulation and the transcriptional level of the tonoplast dicarboxylate transporter *EjtDT1*. Conversely, silencing *EjAGL18* in loquat fruit upregulated the malic acid content and *EjtDT1* expression level. Dual-luciferase assays and yeast one-hybrid experiments further confirmed that EjAGL18 could bind to the promoter of *EjtDT1* and repress its transcriptions. Furthermore, the transient overexpression of *EjtDT1* in loquat fruit increased the malic acid content. These results revealed that EjAGL18 negatively regulates malic acid content by repressing *EjtDT1* in loquat. This study broadens the understanding of the MADS-box TF’s regulatory mechanisms in malic acid and provides new insights into fruit flavor improvement in loquat.

## 1. Introduction

Loquat (*Eriobotrya japonica* Lindl.) is an important Rosaceae fruit tree with high economic value, originating from South China. At present, the commercial cultivation of loquat is widespread across numerous countries, such as China, Japan, Spain, etc. [1]. Loquat is mostly consumed as fresh fruit, making fruit quality the most important index for production and market value. In addition to appearance quality (size, shape, color and absence of defects and decay), more and more consumers are interested in organoleptic quality aspects such as flavor and texture. Fruit acidity is an important indicator of fruit ripening and fruit organoleptic quality and affects fruit flavor along with soluble sugars [2,3]. Elevated fruit acidity, as a primary determinant, diminishes the fruit quality and market value of loquat. Fruit acidity is governed by the concentration and types of organic acids in the vacuoles of fruit cells [4,5]. For most ripe fruits, the dominant organic acids are malic, citric and tartaric acids [6,7,8,9]. In ripe loquat fruit, the content of malic acid accounts for 56–92% of the organic acid content [10], which becomes a major factor causing sour taste. Therefore, elucidating how malic acid accumulates in loquat fruit cell will contribute to the improvement in loquat fruit flavor from the perspective of molecular biology.

Malate, the major form of malic acid in fruit, accumulates in fruit cells through metabolic synthesis and vacuolar storage. The metabolic synthesis of malate mainly depends on the conversion of oxaloacetate (OAA) in the cytosol, the tricarboxylic acid cycle in the mitochondria and the glyoxylate cycle in glyoxysomes [2]. Metabolic enzymes play a crucial role in these processes. Phosphoenolpyruvate carboxylase (PEPC, EC 4.1.1.31), an essential enzyme involved in cytosolic malate synthesis, catalyzes the conversion of phosphoenolpyruvate (PEP), an intermediate of the glycolysis pathway, to OAA [11]. NAD-malate dehydrogenase (NAD-MDH, EC 1.1.1.37) facilitates the reversible conversion between OAA and malate [5,12]. NADP-malic enzyme (NADP-ME, EC 1.1.1.40) promotes the degradation of malate in the cytosol, resulting in malic acid reduction throughout fruit maturation [13]. After malate’s synthesis in the cytosol, it is transported into the vacuoles of fruit cells. The malate transport process mainly relies on transporters and proton pumps, including aluminum-activated malate transporter (ALMT), tonoplast dicarboxylate transporter (tDT), vacuolar H+ -ATPase (V-ATPase), H+ pyrophosphatase (V-PPase) and P-type ATPase proton pumps [14,15,16]. ALMT and tDT are responsible for malate transport into vacuoles, and proton pumps transport protons into vacuoles. The latter process generates proton electrochemical gradients, providing the driving force for malate transport across the tonoplast [15,17]. Finally, malate is protonated and accumulated as the form of malic acid in the vacuoles of fruit cells [2,18].

Transcriptional regulation is an important internal factor of malate accumulation in fruit, in which transcription factors (TFs) play key roles. Numerous TFs have been reported to control malate accumulation through regulating the expression of malate metabolic enzymes and transporter genes, such as MYB, WRKY, bHLH, etc. [15,19,20,21]. Interestingly, malic acid constitutes 85% of the organic acids in apple fruit [21], making it a hot object for studying malate regulatory mechanisms. In apple, MdWRKY126 promoted malate accumulation by activating the expression of *MdMDH5* [20]. MdbHLH3 bound to the promoter of *MdcyMDH* and activated its expression level, thereby enhancing malate accumulation in apple fruit [21]. To date, R2R3-MYB are the most commonly reported TFs that are closely associated with apple fruit acidity. For example, MdMYB1/10 regulated malate accumulation via vacuolar transporters (*MdVHAs*, *MdVHP1* and *MdtDT*) [19]. MdMYB73 was also found to affect malate content by modulating the vacuolar transport system (*MdALMT9*, *MdVHA-A* and *MdVHP1*). Furthermore, MdMYB73 could interact with the MdCIbHLH1 TF and enhance the activity of the above-mentioned target genes [15]. Later, MdMYB123 was found to control malate accumulation via the regulation of *MdMa1* and *MdMa11* [18]. MdMYB44 could downregulate malate content by repressing the expression of *MdMa1/10*, *MdVHA-A3* and *MdVHA-D2* in apple [22]. Similar TF regulatory mechanisms have been found in other fruit crops. In pear, PpWRKY44 could bind to the promoter of *PpALMT9* and activate its transcription, thereby regulating salinity-induced malate accumulation [23]. Another WRKY TF in pear, PbWRKY26, was also found to positively regulate malate content via the activation of *PbMDH3* [24]. In jujube, ZjWRKY7 could bind to the promoter of *ZjALMT4* and activate its expression level, ultimately resulting in malic acid accumulation [25]. In previous studies on loquat, several genes related to fruit ripening and flavor were identified based on transcriptome, metabolome, genome assembly and resequencing analyses [26,27]. However, the transcriptional regulation of malic acid accumulation in loquat fruit is not well understood.

In this study, *EjAGL18*, a *MADS-box* gene, was obtained from seedless triploid loquat ‘Wuhe Zaoyu’, which has excellent flavor with lower malic acid content compared to other loquat cultivars [4,26]. We studied the role of *EjAGL18* in malic acid accumulation in loquat fruit. It was found that EjAGL18 functioned as a transcriptional repressor, suppressing the accumulation of malic acid in loquat fruit. Further biochemical analyses demonstrated that EjAGL18 could bind to the promoter of *EjtDT1*, inhibiting its expression and consequently reducing malic acid content. Our findings provide valuable insights into the molecular regulatory mechanisms of malic acid accumulation and fruit acidity in loquat.

## 2. Results

### 2.1. Identification, Expression Profiling Analysis, Phylogenetic Analysis, Subcellular Localization and Transcriptional Activity Assay of EjAGL18

In our earlier research, we conducted a transcriptome analysis to investigate taste and ripening in diploid and triploid loquat fruits [26]. We found that the expression profiling of the *EjAGL18* (*Eja06G004450*) gene presented different transcription levels between cultivars but the same increasing trend during the G3-G5 stages of loquat fruits (Appendix A). Further real-time quantitative PCR (RT-qPCR) and malic acid determination showed that the expression of *EjAGL18* continuously increased while malic acid content decreased during loquat fruit ripening, suggesting their inverse relationship (Figure 1A–C). Therefore, *EjAGL18* became a candidate gene for further research on malic acid accumulation in loquat.

EjAGL18 encoded a protein of 263 amino acids containing conserved MADS-MEF2-like and K-box domains and polymorphic I and C domains, indicating that EjAGL18 belonged to the MIKC type of MADS-box proteins (Figure 2A). Phylogenetic analysis showed that EjAGL18 shared the highest homology with PbAGL18, PpAGL18 and MdAGL18 and was also a relative of AGL15 proteins (Figure 2B). This suggested that AGL15/18 proteins might have similar biofunctions.

The subcellular localization results showed that the GFP-tagged EjAGL18 fusion protein was localized to the cellular nucleus (Figure 3A). To verify the transcriptional activation of EjAGL18, the Y2HGold yeast strains containing BD-EjAGL18, BD-EjAGL18^1–84^ (containing MADS-MEF2-like domain), BD-EjAGL18^85–263^ (containing I, K, C domain), empty pGBKT7 vector (negative control) and pGBKT7-p53 (positive control) were separately spot-cultured in synthetic dropout media. The results showed that only the positive control yeast strain of pGBKT7-p53 grew normally in SD/-Trp/-His/-Ade plates, while other yeast strains did not grow in the SD/-Trp/-His/-Ade plates (Figure 3B). These results indicated that EjAGL18 was a nucleus-located MADS-box TF without transcriptional activating activity.

### 2.2. Transient Transformation of EjAGL18 Affects Malic Acid Accumulation in Loquat Fruit

To understand the role of EjAGL18 in malic acid accumulation, a pTRV2-EjAGL18 silencing vector and a pCAMBIA2300-EjAGL18 overexpression vector were transiently transformed into ‘Dawuxing’ loquat fruits in the yellow ripening stage (Figure 4A). The expression of *EjAGL18* in transformed fruits showed effective overexpression and silencing (Figure 4B). Malic acid content was then determined in transformed fruits. Our results indicated that the overexpression of *EjAGL18* decreased malic acid content, whereas the silencing of *EjAGL18* increased it (Figure 4C). Therefore, these results demonstrated that *EjAGL18* negatively regulated malic acid accumulation in loquat.

### 2.3. EjAGL18 Binds to the Promoters of EjtDT1 and Inhibits Its Expressions

To investigate the malic acid regulatory mechanisms of EjAGL18, we subsequently focused on the downstream genes related to malate metabolism and transport. Based on our previous transcriptome data [26], other malate-related reports [5] and RT-qPCR results, four genes were found to have high correlation with malic acid, namely *EjNADP-MDH* (*Eja05G022910*), *EjNADP-ME* (*Eja11G016970*), *EjtDT1* (*Eja02G001080*) and *EjtDT2* (*Eja02G000920*) (Appendix A). We further analyzed their expression levels in *EjAGL18* transiently transformed loquat fruits by RT-qPCR. It was found that among these four genes, the expressions of *EjNADP-ME* and *EjtDT1* were significantly downregulated in *EjAGL18*-overexpressing fruits (Figure 5A). This suggested that EjAGL18 could affect malic acid content by downregulating *EjNADP-ME* and *EjtDT1*. However, only the expression level of *EjtDT1* was upregulated in *EjAGL18*-silencing fruits (Figure 5B,C). The downregulation of *EjNADP-ME* in *EjAGL18*-overexpressing fruits and no differential expression changes in *EjNADP-ME* in *EjAGL18*-silencing fruits were probably due to the indirect effects of other stronger EjAGL18-induced factors on the *EjNADP-ME* expression level, rather than the direct inhibition of EjAGL18. Therefore, we proposed the hypothesis that *EjtDT1* might be a downstream target gene of EjAGL18.

To test this hypothesis, yeast one-hybrid (Y1H) and transient dual-luciferase (LUC) assays were performed. The CarG box, CC(A/T)_6_GG or C(A/T)_8_G, has previously been reported as a binding site of the MADS-box TF [28]. Therefore, we further analyzed the promoter (2 kb upstream from the start codon) of *EjtDT1*. After cloning and isolation, we found three CarG boxes in the *EjtDT1* promoter (Figure 5D). The Y1H results showed that the transformants with pAbAi-EjtDT1pro and AD-EjAGL18 normally grew on SD/-Leu (300 ng/mL ABA) plates, while the control containing pAbAi-EjtDT1pro with an empty AD vector did not grow (Figure 5E), indicating the ability of EjAGL18 to bind to the *EjtDT1* promoter in vitro. Notably, although the promoters of *EjNADP-ME/MDH* and *EjtDT2* also contained CarG boxes, the Y1H results showed that there was no binding between EjAGL18 and their promoters (Appendix A), which was consistent with their lack of significant differential expression changes in *EjAGL18* transformed fruits. Therefore, *EjNADP-ME/MDH* and *EjtDT2* were not direct targets of EjAGL18 in this study.

To further investigate whether EjAGL18 directly inhibited the promoter activities of *EjtDT1*, a LUC assay was subsequently performed. pGreenII 62-SK-EjAGL18 (35Spro:: EjAGL18) was used as the effector, and EjtDT1pro::LUC was used as the reporter (Figure 5F). It was found that the promoter activity was significantly inhibited in tobacco leaves coexpressing EjtDT1pro::LUC together with 35Spro::EjAGL18 compared to the control leaves (Figure 5G and Appendix A). These experiments evidenced that EjAGL18 was able to inhibit the transcription of *EjtDT1* by binding to its promoter in vivo.

### 2.4. EjAGL18 Controls Malic Acid Accumulation by Regulating the Expression of EjtDT1 in Loquat

To characterize the biological functions of *EjtDT1*, the pCAMBIA2300-EjtDT1 construct was transiently transformed into loquat fruits (Figure 6A). The RT-qPCR assay showed that our transient transformation significantly overexpressed *EjtDT1* in loquat fruits (Figure 6B). Malic acid content was further determined by HPLC. The results demonstrated that the overexpression of *EjtDT1* increased the malic acid content (Figure 6D), suggesting the key role of *EjtDT1* in the malic acid accumulation of loquat fruit.

To further clarify that the EjAGL18 affected the accumulation of malic acid by directly regulating the expression of *EjtDT1*, the pCAMBIA2300-EjtDT1 recombinant vector and pTRV2-EjAGL18 recombinant vector were transiently co-transformed into loquat fruits, with the empty vectors pCAMBIA2300 and pTRV2 as controls (Figure 6A). The results showed that the overexpression of *EjtDT1* and the simultaneous silencing of *EjAGL18* significantly increased the *EjtDT1* transcriptional level and malic acid content compared to the overexpression of *EjtDT1* alone (Figure 6B–D). These results further confirmed that EjAGL18 could bind to the promoter of *EjtDT1*, repressing its expression and thereby affecting the accumulation of malic acid in loquat fruits.

## 3. Discussion

Malate is an important intermediate of many metabolic processes in plants and plays key roles in the regulation of osmotic pressure, pH homeostasis, nutrient absorption and stress resistance [29,30,31]. More importantly, as the major form of malic acid in fruit cells, malate is a crucial factor affecting fruit flavor and taste, particularly the perception of sweetness [2]. For flesh fruits, consumers prefer fruits with lower acidity. This is because elevated acidity in fruit results in a disagreeable flavor, thus reducing fruit quality and commercial value [4,8,32]. Malic acid is the dominant organic acid in loquat fruit, making it a leading factor influencing loquat fruit acidity [4,5,26]. Therefore, it is necessary to investigate malate accumulation in loquat fruit.

Currently, studies on the acidity of loquat fruit mainly focus on the different organic acid contents and activities of malate metabolic enzymes [4,5,8,26]. Few studies explain the regulatory mechanisms of malate in loquat fruit. In this study, a candidate gene regulating malate in loquat fruit, *EjAGL18* (*Eja06G004450*), was investigated by analyzing the correlation between its transcriptional pattern and the pattern of malic acid accumulation in different developmental stages of ‘Changbai No.1’ (CB_1), ‘Wuhe Zaoyu’ (WHZY) and ‘Huayu Wuhe No.1’ (HYWH) loquat fruits using RT-qPCR and HPLC, respectively. The results showed that the *EjAGL18* transcript level was inversely proportional to malic acid content. So, we hypothesized that *EjAGL18* played a role in the accumulation of malic acid in loquat.

Here, EjAGL18 was found to be a nucleus-located MADS-box transcription factor (TF) without transcriptional activation activity. MADS-box is one of the most extensive TF families in plants [33].The name of MADS is derived from the initials of the *MCM1*, *AGAMOUS*, *DEFICIENS* and *SRF* classes of *MADS-box* genes [34]. In plants, the major type of MADS-box TF is the MIKC-type protein, which contains MADS-box and I-, K- and C-terminal domains [35]. Among these four domains, the MADS-box domain is the most conserved, comprising approximately 50–60 amino acids, and confers DNA binding, nuclear localization and dimerization capabilities to TFs [34,36]. *MADS-box* genes function in almost every developmental process in plants, including floral development [37], flowering time control [38], hormone signal transduction [39], stress response [40] and root, ovule and seed development [41,42,43]. Importantly, MADS-box TFs are the core regulators of fruit development and ripening [44,45]. To date, many *MADS-box* genes have been found to be involved in fruit ripening, such as *RIPENING-INHIBITOR* (*RIN*) [46,47,48], *FRUITFULL1/2* (*FUL1/2*) [49,50,51], *TOMATO AGAMOUS-LIKE 1* (*TAG1*) [52,53,54], etc. Phylogenetic analysis showed that EjAGL18 was the homolog of AtAGL18 in *Arabidopsis* and shared high homology with AGL18 proteins from Rosaceae fruits, including apple, pear and sweet cherry. In addition, the results showed that AGL18 is also a relative of AGL15. These results suggested similar biofunctions between EjAGL18 and other AGL18 or AGL15 proteins. Although AGL15/18 were mostly reported to repress the floral transition [55], affect floral development [56] and promote somatic embryogenesis [57], several AGL15/18 were also found to affect fruit ripening and texture. For example, *CpAGL18* was found to positively regulate fruit ripening in papaya [58]. Ge et al. [59] demonstrated the important role of *EjAGL15* in regulating lignin accumulation in loquat fruit. However, studies on MADS-box TFs controlling fruit flavor, particularly fruit acidity, are limited.

To analyze the role of *EjAGL18* in loquat fruit acidity, the transient transformation of *EjAGL18* in loquat was performed. We found that the overexpression of *EjAGL18* in loquat fruit decreased malic acid content, whereas the silencing of *EjAGL18* increased it. Malic acid accounts for a large proportion of organic acids that affect fruit flavor and acidity and can also be used as a critical indicator of fruit acidity and fruit ripening [60]. In loquat, variations in malic acid content are directly responsible for variations in titratable acidity [5], which is a common indicator of fruit acidity. Therefore, we believed that *EjAGL18* affected loquat fruit acidity by controlling malic acid accumulation. To explain how *EjAGL18* controls malic acid and considering the key influences of malate metabolism and transport on malic acid accumulation, we paid more attention to malate metabolic enzyme genes and malate-related transporter genes. In combination with our previous transcriptome data [26] and related studies [2,5], an NADP-malate dehydrogenase gene (*EjNADP-MDH*, *Eja05G022910*), an NADP-malic enzyme gene (*EjNADP-ME*, *Eja11G016970*) and two tonoplast dicarboxylate transporters (*EjtDT1/2*, *Eja02G001080*/*Eja02G000920*) were screened as candidate EjAGL18 downstream target genes. Due to the lack of transcriptional activation activity of EjAGL18, we focused on genes that were downregulated in *EjAGL18*-overexpressing fruits. The results showed that *EjNADP-ME* and *EjtDT1* were both significantly downregulated in *EjAGL18*-overexpressing fruits, but only *EjtDT1* was significantly upregulated in *EjAGL18*-silencing fruits. Next, we found that the promoter region of *EjtDT1* contained three CarG boxes, known binding sites for MADS-box TFs [28], suggesting that it might bind to EjAGL18. To verify this possibility, yeast one-hybrid (Y1H) and transient dual-luciferase (LUC) assays were performed. The results illustrated that EjAGL18, as a transcriptional repressor, bound to the promoter of *EjtDT1*, resulting in the repression of its promoter activity and gene expression level. The lack of binding in the Y1H results between the *EjNADP-ME* promoter and EjAGL18 suggested that the negative response of *EjNADP-ME* to *EjAGL18* overexpression was probably due to the indirect effects of other more potent EjAGL18-induced factors rather than the direct regulation of EjAGL18. Normally, transcriptional repressors bind either directly or indirectly to DNA, and they can regulate transcription from binding sites proximal to, or at a distance from, the promoter [61]. Here, the direct binding of EjAGL18 and the *EjtDT1* promoter prevented the competitive binding of other TFs to target the DNA binding sites of the *EjtDT1* promoter. Similar transcriptional repression mechanisms were also found in many TFs, such as ClNAC68 in watermelon [62], VvBBX44 in grape [63] and MaC2H2-1/2 in banana [64].

The tonoplast dicarboxylate transporter (tDT) is a major malate transporter responsible for transport between the cytoplasm and the central vacuole and plays a key role in malate accumulation [14]. In *Arabidopsis*, the absence of *AttDT* in mutants had a correlation with the impaired accumulation of malate and fumarate in leaves [14,65]. In tomato, *SlTDT* positively regulated malate content but negatively regulated citrate content [66]. In the halophyte *Spartina alterniflora*, the overexpression of *SaTDT* affected the accumulation of malic and citric acids in cells [67]. A similar function of *EjtDT1* was found in this study. The overexpression of *EjtDT1* promoted the accumulation of malic acid in loquat fruits. When *EjAGL18* was silenced in *EjtDT1*-overexpressing fruits, the content of malic acid was significantly higher than that in fruits that overexpressed *EjtDT1* alone. Therefore, EjAGL18 negatively regulated malic acid accumulation by repressing the transcription of *EjtDT1*, thereby affecting the acidity of loquat fruit.

Based on this, a regulatory model of EjAGL18 regulating malic acid accumulation was crafted to illustrate our work (Figure 7). Malate could be transported from the cytosol into the vacuole by the transporter EjtDT1. The overexpression of *EjAGL18* repressed the transcriptional level of *EjtDT1* by binding to its promoter, thereby affecting the transport of malate, resulting in malate accumulation in the cytosol and inhibiting the upstream synthesis process of malate. The decreasing malate^2–^ and H^+^ combined and formed less malic acid in the vacuole, reducing the acidity of loquat fruit.

## 4. Materials and Methods

### 4.1. Plant Materials and Growth Conditions

Loquat fruits were collected at 21, 23 and 26 weeks post-anthesis (WPA), namely the swollen fruit stage (G3), breaker stage (G4) and ripe stage (G5), from the diploid cultivar ‘Changbai 1’ (CB_1) and the triploid cultivars ‘Huayu Wuhe 1’ (HYWH) and ‘Wuhe Zaoyu’ (WHZY) [26]. These collected fruits were used for a reverse transcription quantitative PCR (RT-qPCR) assay and the determination of malic acid. Plants were grown naturally at the plantation base of Southwest University, Chongqing, China (29°80′ N, 106°40′ E). For each replicate, at least six loquat fruits were collected from three trees under similar growing conditions. ‘Dawuxing’ loquat fruits were used for the transient transformation of gene silencing and overexpression and were planted under natural conditions at the loquat planting base, Mengzi, Yunnan, China (23°22′ N, 103°20′ E). The sampled fruits were promptly flash-frozen in liquid nitrogen and stored at −80 °C for subsequent experiments. Tobacco (*Nicotiana benthamiana*) was cultivated under a photoperiod of 16 h of light and 8 h of darkness at 23 °C.

### 4.2. RNA Extraction and RT-qPCR Analysis

For the extraction of total RNA, we used an RNAprep Pure Plant Plus Kit (Polysaccharide- and Polyphenolic-rich) (TIANGEN, Beijing, China). RNA purity, quality and concentration were detected with 1% RNase-free agarose gel electrophoresis and a 2100 Bioanalyzer (Agilent Technologies, Santa Clara, CA, USA). For the synthesis of cDNA, we used a PrimeScript™ RT reagent Kit with gDNA Eraser (TaKaRa, Shiga, Japan). RT-qPCR was performed using NovoStart^®^SYBR qPCR SuperMix plus (E096-01A, Novoprotein, Suzhou, China) on qTOWER^3^ G (Analytik Jena, Jena, Germany). A reaction mixture of 10 µL included 5 µL SYBR mix, 0.2 µL gene-specific forward and reverse primers (10 µmol/L), 1 µL cDNA template (10 ng/µL) and 3.6 µL sterile water. The cycling procedure was as follows: 94 °C for 20 s, 94 °C for 10 s, 58 °C for 10 s and 72 °C for 10s, for 40 cycles. Gene-specific primers were designed using Primer 5.0 software. All primer sequences for RT-qPCR are listed in Appendix A. The relative expression levels of each biological replicate, including three technical replicates, were normalized to the reference gene (qEjactin-F/R, JN004223) [68]. Gene expression levels were calculated and analyzed using the 2^−ΔΔCt^ method [69].

### 4.3. Isolation of EjAGL18, Phylogenetic Analysis and Subcellular Localization of Its Protein

The *EjAGL18* gene was isolated from the triploid loquat ‘WHZY’. Multiple sequence alignment was performed using DNAMAN software (V.6.0) (Lynnon Biosoft, San Ramon, CA, USA). The conserved domains of EjAGL18 were analyzed in NCBI-CDD (https://www.ncbi.nlm.nih.gov/Structure/cdd/wrpsb.cgi (accessed on 20 November 2024)). A phylogenetic tree of EjAGL18 was established using MEGA software (v11.0) via the neighbor-joining method with 1000 bootstrap replicates [70]. The full length of *EjAGL18* without the stop codon was inserted into the modified pCAMBIA2300 vector with the CaMV35S promoter and an eGFP (enhanced green fluorescent protein) tag, forming the fusion expression vector 35S::EjAGL18-GFP. The VirD2N LS-mCherry vector was used as a nucleus marker and was co-transformed with the constructed fusion vector or empty vector into tobacco (*Nicotiana benthamiana*) leaves. Fluorescence signal observation was performed on a confocal laser scanning microscope (LSM 780; Carl Zeiss, Oberkochen, Germany) after the 36 h dark treatment of the transformation. Each transient expression observation was repeated more than three times. The specific primers are listed in Appendix A.

### 4.4. Transcriptional Activation Assay of EjAGL18

The full-length, N-terminal (containing MADS-MEF2-like domain) and C-terminal (containing I, K, C domains) CDSs of *EjAGL18* were each inserted into the pGBKT7 (BD) vector and then transformed into the yeast strain Y2HGold. The empty BD vector was used as a negative control and pGBKT7-p53 as a positive control. Each transformed strain was spot-cultured on the SD/-Trp and SD/-Trp/-His/-Ade plates. Transcriptional activity was finally assessed according to the growth statuses of each strain.

### 4.5. Transient Gene Overexpressing and Silencing Analysis in Loquat Fruit

Gene silencing was induced by *tobacco rattle virus* (TRV). The partial antisense sequence of *EjAGL18* was inserted into the pTRV2 vector. Fusion vectors for gene overexpression were constructed using pCAMBIA2300 with the 35S promoter and the full-length cDNAs of *EjAGL18* and *EjtDT1*. The primer sequences are listed in Appendix A. Each fusion vector was transformed into the *Agrobacterium* strain GV3101 prior to the transient transformation of loquat fruits. The pCAMBIA2300 recombinant vectors were used individually, while the pTRV2 recombinant vectors were used together with pTRV1 at a ratio of 1:1. The empty pCAMBIA2300 and pTRV2 vectors were used as negative controls. Verified cells were expanded and then resuspended in infection buffer (containing MES 10 mmol/L, MgCl_2_ 10 mmol/L, acetosyringone 200 µmol/L) until OD_600_ reached 0.4–0.6, and then they were incubated at 28 °C for 3–4 h before injection [71]. For gene overexpression and silencing injections, ‘Dawuxing’ loquat fruits of uniform size and at the yellow ripening stage (already yellowed but not yet softened) were used. Three biological replicates (represented as 1#, 2# and 3# in Figure 4) were set up, and each biological replicate consisted of six fruits. For each injection, approximately 300 µL of infection buffer was administered using a sterile syringe. Post-injection, the fruits were initially kept in the dark for 24 h, followed by 4 days in natural light at 23 °C. On the fifth day of injection, the injected parts of the loquat fruits were sampled and instantly frozen in liquid nitrogen and then stored at −80 °C. These collected samples were used for the subsequent RT-qPCR and malic acid determination.

### 4.6. Determination of Malic Acid in Loquat Fruit

Malic acids were extracted and determined as described by [4]. An LC-20AT model HPLC system (Shimadzu Co., Kyoto, Japan) with a diode array detector (DAD, SPD-M20A, Kyoto, Japan) and a C_18_ column (5 μm, 250 mm × 4.6 mm, GL Science, Japan) were used for the determination of malic acid. Elution was performed at 30 °C under a mobile phase of 2% KH_2_PO_4_ (pH 2.52)/methanol (v:v) = 9:1, at a flow rate of 0.8 mL/min. The chromatogram was continuously monitored at 210 nm throughout the elution. The injection volume was 10 µL. Malic acid content was calculated from the peak area by an analytical interpolation of the calibration curve based on an external standard method and presented as mg/g of fresh weight (FW). The malic acid content in each biological replicate was determined three times.

### 4.7. Yeast One-Hybrid (YIH) Assay

The promoter region of *EjtDT1* was inserted into the pAbAi vector (pAbAi-EjtDT1pro). The *EjAGL18* coding sequence was inserted into the pGADT7 (AD) vector (AD-EjAGL18). The recombinant vector pAbAi-EjtDT1pro was co-transformed with AD-EjAGL18 into Y1HGold yeast cells. The yeast strain containing pAbAi-EjtDT1pro and an AD empty vector was used as a control. Aureobasidin A (ABA) (Solarbio, Beijing, China) was used to screen the bait yeast strains. Whether EjAGL18 bound to the promoter of *EjtDT1* was finally assessed by the growth statuses of strains. The primer sequences for the Y1H experiment are listed in Appendix A.

### 4.8. Dual-Luciferase Reporter Assay

The complete coding sequence of *EjAGL18* was inserted into the pGreenII 62-SK vector with the 35S promoter (35Spro::EjAGL18) as the effector. The promoter of *EjtDT1* was inserted into the pGreenII 0800-LUC vector as the reporter (EjtDT1pro::LUC). The pGreenII 62-SK empty vector was used as a negative control. After 48–72 h of the *Agrobacterium*-mediated (GV3101) infection of the effector and reporter vectors into tobacco leaves, luciferase activities were determined using the Promega Dual-Luciferase^®^ Reporter Assay System Kit (E1910) on a microplate reader (Varioskan Flash 3001, THERMO Fisher Scientific, Waltham, MA, USA). Finally, the relative ratio of firefly luciferase (LUC)/renilla luciferase (REN) activity was used to analyze the transcriptional activity of the promoter. Each analysis was repeated three times. All primer sequences are listed in Appendix A.

### 4.9. Statistical Analysis

IBM SPSS Statistics 27 was used for significance analyses. Student’s *t*-test was used to analyze the significance between two comparisons, indicated as * for *p* < 0.05, ** for *p* < 0.01 and *** for *p* < 0.001. An ANOVA was performed using Duncan’s test to analyze the significance level of multiple comparisons, and different lowercase letters were used to indicate statistically significant differences (*p* < 0.05).

## 5. Conclusions

In this study, *EjAGL18*, a *MADS-box* gene, was isolated from the triploid loquat ‘Wuhe Zaoyu’, whose expression pattern presented an opposite trend to malic acid accumulation in loquat fruit. EjAGL18 is a nucleus-located MADS-box transcription factor without transcriptional activation activity. The overexpression and silencing of *EjAGL18* revealed its negative control of malic acid in loquat fruit. Yeast one-hybrid and dual-luciferase assays demonstrated that EjAGL18 could bind to the promoter of the tonoplast dicarboxylate transporter gene *EjtDT1* and inhibit its expression. The overexpression of *EjtDT1* increased the malic acid content in loquat fruit. Therefore, EjAGL18 negatively regulated malic acid accumulation by repressing the expression of *EjtDT1* and played a key role in the fruit acidity of loquat. This study provides new insights into the regulatory mechanisms of the MADS-box TF in terms of malic acid and the improvement in fruit flavor in loquat.

## Figures and Tables

**Figure 1 ijms-26-00530-f001:**
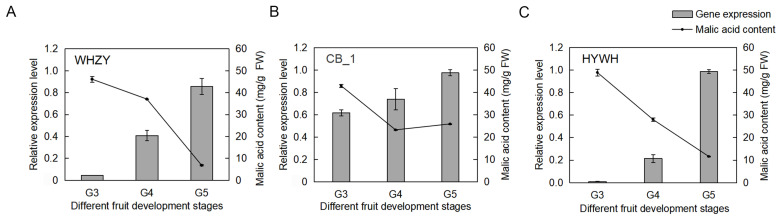
The expression pattern of *EjAGL18* in ‘Wuhe Zaoyu’ (**A**), ‘Changbai No.1′ (**B**) and ‘Huayu Wuhe No.1’ (**C**) loquat fruits in different development stages. ‘WHZY’, ‘HYWH’ and ‘CB_1′ represent the triploid loquat cultivars ‘Wuhe Zaoyu’ and ‘Huayu Wuhe No.1’ and the diploid loquat cultivar ‘Changbai No.1’, respectively. G3-G5 represent three development stages of loquat fruits, namely 21, 23 and 26 weeks post-anthesis. Gene expression is shown as a bar chart, and malic acid content is shown as a line graph. The values are shown as the mean ± SE.

**Figure 2 ijms-26-00530-f002:**
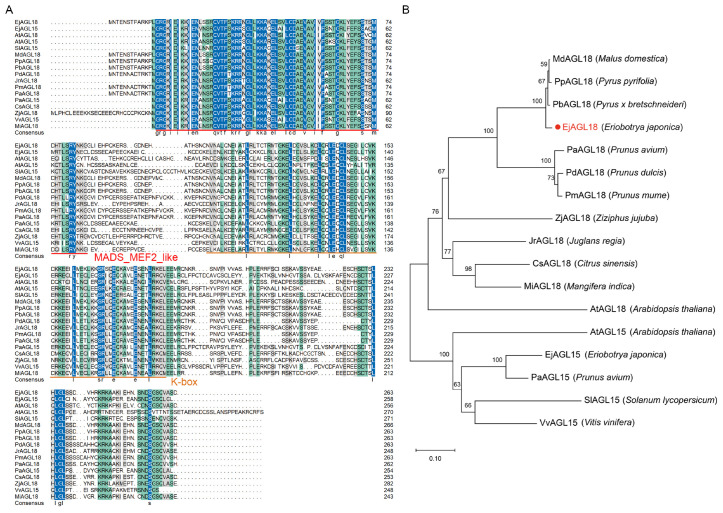
The multiple sequence alignment (**A**) and phylogenetic analysis of EjAGL18 (**B**). A total of 17 AGL proteins were used for multiple sequence alignment analysis and phylogenetic tree construction. The bootstrap values are shown near the branch in black text. The GenBank accession numbers of these homologous proteins are as follows: AtAGL15, NP_001330207.1; AtAGL18, OAP03427.1; CsAGL18, XP_024949072.2; EjAGL15, WGN96212.1; JrAGL18, XP_018818423.1; MdAGL18, XP_008351603.2; MiAGL18, XP_044468827.1; PaAGL15, XP_021821687.1; PaAGL18, XP_021826391.1; PbAGL18, XP_009363197.2; PdAGL18, XP_034225225.1; PmAGL18, XP_008244251.1; PpAGL18, AJW29031.1; SlAGL15, XP_004229674.1; VvAGL15, AUJ18468.1; ZjAGL18, XP_048321544.2. The sequences of MADS-MEF2-like and K-box were underlined in red and orange, respectively.

**Figure 3 ijms-26-00530-f003:**
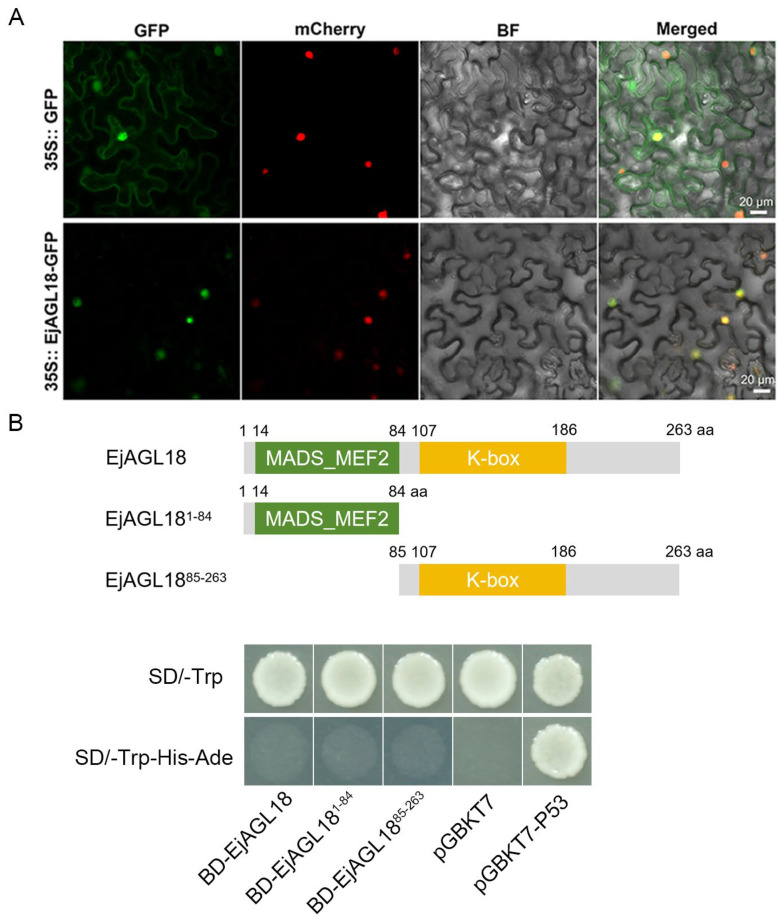
Subcellular localization and transcriptional activation activity assay of EjAGL18. (**A**) Subcellular localization of EjAGL18. GFP, GFP fluorescence channel; mCherry, red fluorescence channel; BF, bright field; Merged, merged image of GFP, red and bright field. (**B**) Transcriptional activity assay of EjAGL18 in yeast.

**Figure 4 ijms-26-00530-f004:**
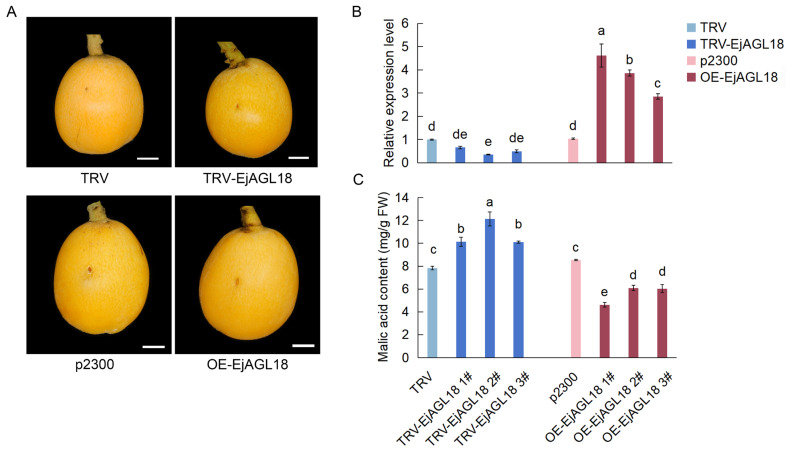
Phenotype, expression level changes in *EjAGL18* and malic acid content in *EjAGL18* transient transformation loquat fruits. (**A**) Phenotypes of *EjAGL18* transiently transformed fruits. TRV-EjAGL18 and OE-EjAGL18 represent silencing and overexpression of *EjAGL18*, respectively. Empty vectors TRV1 and TRV2 (TRV) and pCAMBIA2300 (p2300) served as control. Scale bar, 1 cm. (**B**) Expression level changes in *EjAGL18* in overexpressing and silencing fruit. (**C**) Content changes in malic acid in *EjAGL18* transformed fruit. Here, 1#, 2# and 3# represent three biological replicates of *EjAGL18* transformation. Values are shown as mean ± SE. Letters on each bar represent significant differences calculated by one-way ANOVA at *p* < 0.05.

**Figure 5 ijms-26-00530-f005:**
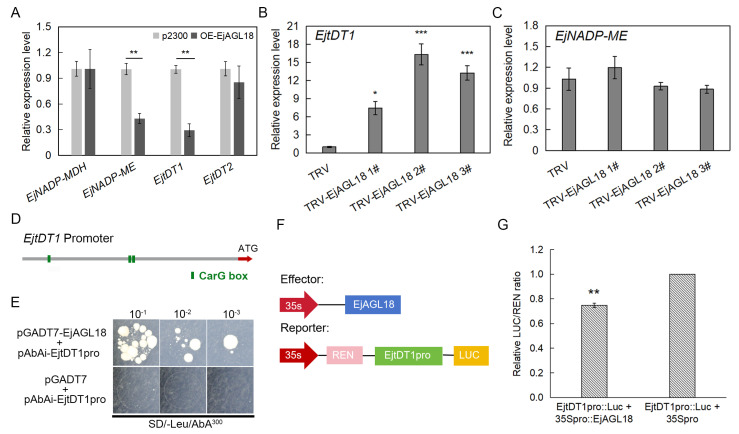
EjAGL18 binds to the promoter of *EjtDT1*. (**A**) The expression of *EjNADP-MDH*, *EjNADP-ME*, *EjtDT1* and *EjtDT2* in *EjAGL18*-overexpressing loquat flesh. (**B**) The expression of *EjtDT1* in *EjAGL18*-silencing loquat flesh. (**C**) The expression of *EjNADP-ME* in *EjAGL18*-silencing loquat flesh. TRV-EjAGL18 and OE-EjAGL18 represent the silencing and overexpression of *EjAGL18*, respectively. TRV and p2300 represent the controls of silencing and overexpression, respectively. (**D**) The CarG box sites in the promoter region of *EjtDT1.* (**E**) Y1H assay. The co-transformation yeast cells of the empty vector pGADT7 and the *EjtDT1* promoter (pGADT7 + pAbAi-EjtDT1pro) were used as controls. The screened concentration of Aureobasidin A (ABA) was 300 ng/mL. (**F**) A schematic diagram of the effector vector (35Spro::EjAGL18) and LUC reporter vector (EjtDT1pro::LUC). (**G**) The relative ratio of LUC/REN in dual-luciferase assays, and the fluorescence intensity of EjtDT1pro::LUC + 35Spro was set as 1. Values are shown as the mean ± SE from three replicates. The asterisk indicates significance compared to the control using Student’s *t*-test (* *p* < 0.05; ** *p* < 0.01, *** *p* < 0.001).

**Figure 6 ijms-26-00530-f006:**
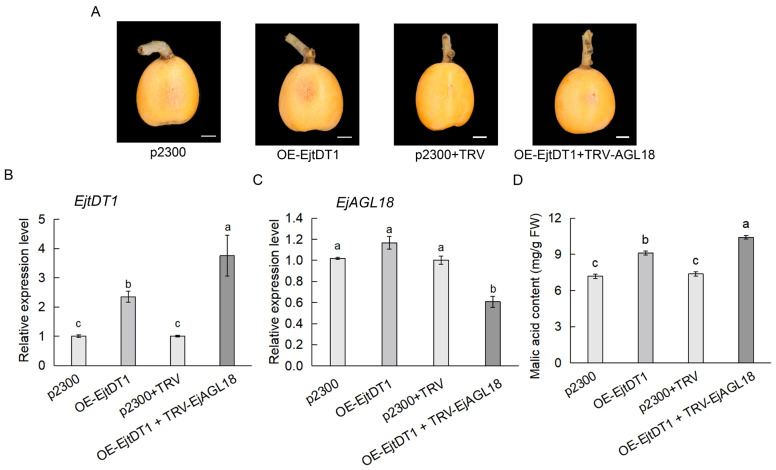
EjAGL18 controls malic acid accumulation by directly regulating the expression of *EjtDT1.* (**A**) The phenotypes of transiently transformed loquat fruits. OE-EjtDT1 represents the overexpression of *EjtDT1*. TRV-EjAGL18 represents the silencing of *EjAGL18*. The empty vectors TRV1 and TRV2 (TRV) and pCAMBIA2300 (p2300) served as the control. Scale bar, 1 cm. (**B**) The mRNA level of *EjtDT1* in transformed fruits. (**C**) The mRNA level of *EjAGL18* in transformed fruits. (**D**) The malic acid content in transformed fruits. Values are presented as the mean ± SE. Letters on each bar represent significant differences that were calculated by a one-way ANOVA at *p* < 0.05.

**Figure 7 ijms-26-00530-f007:**
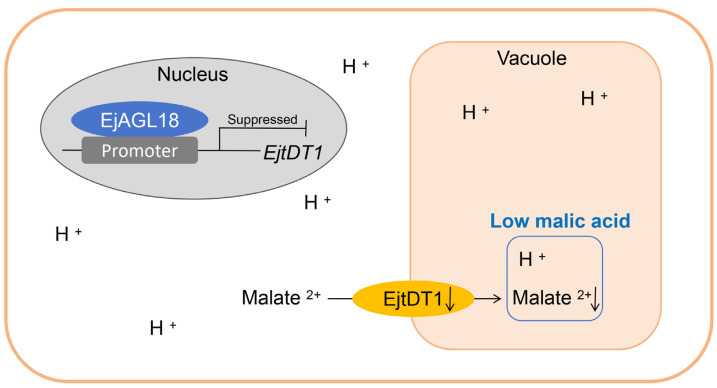
A regulatory model of EjAGL18 repressing malic acid accumulation in loquat fruit. EjAGL18 inhibits the transcriptional level of *EjtDT1*, affecting the transport of malate from the cytosol into the vacuole, leading to the reduced accumulation of malic acid in the vacuole.

## Data Availability

Data are contained within this article.

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
