# Peer review of "The MADS-Box Transcription Factor EjAGL18 Negatively Regulates Malic Acid Content in Loquat by Repressing EjtDT1"

_ijms, 2025, doi:10.3390/ijms26020530_

Round 1

Reviewer 1 Report

Comments and Suggestions for Authors

In the manuscript named “The MADS-box Transcription Factor EjAGL18 Negatively Regulates Malic Acid Content in Loquat by Repressing EjtDT1”, Zhuoheng Chi et al have comprehensively performed molecular experiments to illustrate EjAGL18 function in malic acid metabolism, including qRT-PCR, transient overexpression, and yeast one-hybrid (Y1H), etc. Their results have well demonstrated its function in plant fruit growth process, which would be helpful for genetic breeding in future. However, there were some comments about it.

(1) Authors have displayed EjAGL18 with negatively regulation function in malic acid metabolism, its function was working by inhibiting EjtDT1 expression. The hypothesis was not clear in this manuscript. Authors have selected four genes based previous research, only one gene, EjtDT1, was negative to EjAGL18, while EjNADP-ME was not responsive in silencing experiment? Why? If the EjAGL18 works as TFs, binding to cis-elements in promoters, it would bind to many downstream genes, why one gene EjtDT1? Why EjtDT2 didn’t work. Authors didn’t analyze cis-elements distribution in other function genes. In summary, there was no strong evidence to support their hypothesis, more experiments would be needed.

(2) Dual-luciferase assays original results would be listed in supplements.

(3) Some analysis have missed Duncans test, see figure 6B and 6C, “letters on each bar”, 6A has also missed scale bar.

(4) In multiple sequence alignment of AGL18 genes, authors have selected many supposed members, started with “XP”, see figure 2, why did not select some public members in published reports?

(5) Authors have also described some refs, including AGL18, and tDT function in malate accumulation, but they have also missed their linking evidence. Especially, EjAGL18 would bind to EjtDT1 promoter, which would prevent another TFs binding to EjtDT1, authors need more refs to explore this point of view.

Author Response

Dear reviewer,

Thank you very much for taking the time to review this manuscript. Your comments were important for us to improve our manuscript. We have studied your comments carefully and revised manuscript according to these comments. We hope our revised manuscript could meet your requirements. Please find the detailed responses below and the corresponding revisions highlighted using red font in the re-submitted files.

Comment 1: In the manuscript named “The MADS-box Transcription Factor EjAGL18 Negatively Regulates Malic Acid Content in Loquat by Repressing EjtDT1”, Zhuoheng Chi et al have comprehensively performed molecular experiments to illustrate EjAGL18 function in malic acid metabolism, including qRT-PCR, transient overexpression, and yeast one-hybrid (Y1H), etc. Their results have well demonstrated its function in plant fruit growth process, which would be helpful for genetic breeding in future. However, there were some comments about it.

Response 1: Thanks for your positive affirmation and evaluation for our study. According to your comments, we have carefully revised the manuscript and the content of amendments was also highlighted using red font in our revised manuscript. The detailed point-by-point response to your comments is as follows.

Comment 2: Authors have displayed EjAGL18 with negatively regulation function in malic acid metabolism, its function was working by inhibiting EjtDT1 expression. The hypothesis was not clear in this manuscript.

Response 2: Thanks for your comments. We totally understand your doubt and fully agree with this comment. To clarify the hypothesis that EjAGL18 binds to EjtDT1 instead of other candidate genes, we have added the RT-qPCR results, distribution of CarG box in the promoters and previous obtained yeast one-hybrid (Y1H) results of other three candidate downstream genes (EjNADP-ME/MDH and EjtDT2) in Figure S2 and Figure S3. In fact, we have conducted yeast one-hybrid experiments of EjNADP-ME/MDH and EjtDTT1/2 promoters with EjAGL18 at the same time. But we only displayed the binding results of EjtDT1. We apologize for the doubts caused by incomplete experiment results and have added other Y1H results of no binding to EjAGL18 in Figure S3. Meanwhile, we have added following content in section 2.3 (L193): “Notably, although the promoters of EjNADP-ME/MDH and EjtDT2 also contained CarG boxes, the Y1H results showed that there was no binding between EjAGL18 and their promoters (Figure S3), which was consistent with the their no significant differential expression changes in EjAGL18 transformed fruits. Therefore, EjNADP-ME/MDH and EjtDT2 were not the direct targets of EjAGL18 in this study.”

Comment 3: Authors have selected four genes based previous research, only one gene, EjtDT1, was negative to EjAGL18, while EjNADP-ME was not responsive in silencing experiment? Why?

Response 3: In this manuscript, EjNADP-ME and EjtDT1 were both downregulated in EjAGL18 overexpressing fruits. However, EjNADP-ME was not responsive in EjAGL18 silencing fruits. We speculate that there might be other, more potent factors induced by the overexpression of EjAGL18 that affect the expression of EjNADP-ME,rather than the direct regulation of EjAGL18. But this remained to be further investigated in the future. To avoid misunderstandings, we have made following additions and modifications in section 2.3 (L177): “This suggested that EjAGL18 could affect malic acid content by downregulating EjNADP-ME and EjtDT1. However, only the expression level of EjtDT1 was upregulated in EjAGL18 silencing fruits (Figure 5B, C). The downregulation of EjNADP-ME in EjAGL18 overexpressing fruits and no differential expression changes of EjNADP-ME in EjAGL18 silencing fruits were probably due to the indirect effects of other stronger EjAGL18-induced factors on EjNADP-ME expression level, rather than the direct inhibition of EjAGL18. Therefore, we proposed the hypothesis that EjtDT1 might be a downstream target gene of EjAGL18.” In discussion (L309), we also added “The lack of binding of Y1H result between the EjNADP-ME promoter and EjAGL18 suggested that the negative response of EjNADP-ME to EjAGL18 overexpression was probably due to the indirect effects of other, more potent EjAGL18-induced factors rather than the direct regulation of EjAGL18.”

Comment 4: If the EjAGL18 works as TFs, binding to cis-elements in promoters, it would bind to many downstream genes, why one gene EjtDT1? Why EjtDT2 didn’t work.

Response 4: Thanks for your question. Although EjtDT1 and EjtDT2 both encoded the tonoplast dicarboxylate transporters, their localizations on chromosomes and promoter sequences were totally different. According to our genome data, EjtDT1 is localized on the LG01 chromosome while EjtDT2 is localized on the LG02 chromosome in loquat. The promoter of EjtDT1 contains three CarG boxes, whereas the promoter of EjtDT2 contains only one CarG box. The additional Y1H experiment result showed that there was no binding between EjAGL18 and the EjtDT2 promoter, suggesting that EjtDT2 was not direct target of EjAGL18. Accordingly, the Y1H result and distribution of CarG box in EjtDT2 promoter has been added to Figure S3. There are many reasons why EjAGL18 does not bind to the EjtDT2 promoter, including the oligomerization state of EjAGL18 and the influence of the CarG box flanking sequence. More comprehensive research is needed to investigate the specific reasons for the lack of integration between EjAGL18 and other downstream genes in the future.

Comment 5: Authors didn’t analyze cis-elements distribution in other function genes.

Response 5: Thanks for your advice. As the CarG box is the known binding site of MADS-box TF, we have added the distribution of CarG box in other candidate downstream genes in Figure S3.

Comment 6: In summary, there was no strong evidence to support their hypothesis, more experiments would be needed.

Response 6: Thanks for your instructive suggestion. We are fully aware that our manuscript will benefit from more supporting experiments. Therefore, we have supplemented the RT-qPCR of downstream genes (EjNADP-ME/MDH and EjtDT1/2) in loquat fruits at different developmental stages in Figure S2, the CarG box distribution promoters and Y1H results of EjNADP-ME/MDH and EjtDT2 in Figure S3. Furthermore, to make our conclusion more robust, the co-transformation of the pCAMBIA2300-EjtDT1 vector with the pTRV2-EjAGL18 vector was additionally conducted in loquat fruits. The results showed that overexpressing EjtDT1 and silencing EjAGL18 at the same time significantly increased the EjtDT1 transcriptional level and malic acid content compared to overexpressing EjtDT1 alone. We believed that RT-qPCR, Y1H, Dual-LUC and transient individual- and co-transformations of EjAGL18 and EjtDT1 were sufficient to support our hypothesis that EjAGL18 could bind to the promoter of EjtDT1, repressing its expression and thereby affecting the accumulation of malic acid in loquat fruits.

Accordingly, the supplementary co-transformation results were updated in Figure 6 and content in section 2.4. The added content is as follows (L224). “To further clarify that the EjAGL18 affected the accumulation of malic acid by direct regulating the expression of EjtDT1, the pCAMBIA2300-EjtDT1 recombinant vector and pTRV2-EjAGL18 recombinant vector were transiently co-transformed into loquat fruits, with the empty vectors pCAMBIA2300 and pTRV2 as controls (Figure 6A). The results showed that the overexpression of EjtDT1 and simultaneous silencing of EjAGL18 significantly increased the EjtDT1 transcriptional level and malic acid content compared to overexpression of EjtDT1 alone (Figure 6B-D). These results further confirmed that EjAGL18 could bind to the promoter of EjtDT1, repressing its expression and thereby affecting the accumulation of malic acid in loquat fruits.” In discussion (L327), we have added “When EjAGL18 was silenced in EjtDT1-overexpressing fruits, the content of malic acid was significantly higher than in fruits that overexpressing EjtDT1 alone.”

Comment 7: Dual-luciferase assays original results would be listed in supplements.

Response 7: Thanks for your kind reminder. We have uploaded the original results of dual- luciferase assays in supplementary Table S2.

Comment 8: Some analysis have missed Duncan’s test, see figure 6B and 6C, “letters on each bar”, 6A has also missed scale bar.

Response 8: Thank you for reminding us of this. We have added the results of Duncan’s test in Figure 6B-D and the scale bar in Figure 6A accordingly.

Comment 9: In multiple sequence alignment of AGL18 genes, authors have selected many supposed members, started with “XP”, see figure 2, why did not select some public members in published reports?

Response 9: Thanks for your question. EjAGL18 was a homology to AGAMOUS-like 18 (AGL18) proteins and was a relative of AGAMOUS-like 15 (AGL15) proteins. AGL15/18 proteins are rarely identified in fruit trees, but are mostly studied and reported in Arabidopsis thaliana and Brassica juncea. To better present the sequence conservation and phylogenetic relationships of EjAGL18 in fruit trees and model plants, we can only select the putative AGL15/18 protein sequences on NCBI. Many reports also use the supposed members to carry out multiple sequence alignment, such as Zhang et al. (2022), Su et al. (2021) and Li et al. (2023). Therefore, we believe that putative protein sequences can also be used for analysis when there are no more published members to choose from.

Zhang, L.H., Z., Ma, B.Q., Wang, C.Z., et al. (2022). MdWRKY126 modulates malate accumulation in apple fruit by regulating cytosolic malate dehydrogenase (MdMDH5). Plant Physiology, 188, 2059-2072.

Su, W.B., Zhang, L., Jiang Y.Y., et al. (2021). EjFWLs are repressors of cell division during early fruit morphogenesis of loquat. Scientia Horticulturae, 287, 110261.

Li, F.F., Fu, M.J., Zhou, S.G., et al. (2023). A tomato HD-zip I transcription factor, VAHOX1, acts as a negative regulator of fruit ripening. Horticulture Research, 10,140-151.

Comment 10: Authors have also described some refs, including AGL18, and tDT function in malate accumulation, but they have also missed their linking evidence. Especially, EjAGL18 would bind to EjtDT1 promoter, which would prevent another TFs binding to EjtDT1, authors need more refs to explore this point of view.

Response 10: Thanks for your instructive advice. We have added discussion and references about transcriptional repression mechanism of EjAGL18 and EjtDT1. In discussion (L312), we have added “Normally, transcriptional repressors bind either directly or indirectly to DNA and they can regulate transcription from binding sites proximal to, or at a distance from, the promoter [61]. Here, the direct binding of EjAGL18 and EjtDT1 promoter prevented competitive binding of other TFs to target DNA binding sites of EjtDT1 promoter. Similar transcriptional repression mechanisms were also found in many TFs, such as ClNAC68 in watermelon [62], VvBBX44 in grape [63], MaC2H2-1/2 in banana [64].”

The added references are listed below.

  1. Gaston, K.; Jayaraman, P.S. Transcriptional repression in eukaryotes: repressors and repression mechanisms. Cell. Mol. Life Sci. 2003, 60, 721-741.
  2. Wang, J.F.; Wang, Y.P.; Zhang, J.; Ren, Y.; Li, M.Y.; Tian S.W.; Yu, Y.T.; Zuo, Y.; Gong, G.Y.; Zhang, H.Y.; et al. The NAC transcription factor ClNAC68 positively regulates sugar content and seed development in watermelon by repressing ClINV and ClGH3.6. Hortic. Res. 2023, 8,214.
  3. Liu, W.W.; Mu, H.Y.; Yuan L.; Li, Y.; Li, Y.T.; Li, S.C.; Ren, C.; Duan, W.; Fan, P.G.; Dai, Z.W.; et al. VvBBX44 and VvMYBA1 form a regulatory feedback loop to balance anthocyanin biosynthesis in grape. Hortic. Res. 2023, 10, uhad176.
  4. Han, Y.C.; Fu, C.C.; Kuang, J.F.; Chen, J.Y.; Wang, J.L. Two banana fruit ripening-related C2H2 zinc finger proteins are transcriptional repressors of ethylene biosynthetic genes. Postharvest Biol. Technol. 2016, 116, 8-15. 

Reviewer 2 Report

Comments and Suggestions for Authors

Authors identified a MADS-box transcription factor, EjAGL18, regulating the malic acid accumulation in loquat. Authors confirmed the nuclear localization of EjAGL18, however, the Y2H-based test showed the lack of trans-activation properties. The expression of EjAGL18 is inversely correlated with the malic acid concentration durig fruit ripening. Dual luciferase and Y1H assays showed that EjAGL18 binds to EjtDT1 promoter and repress its activity. Moreover, the transient overexpression of EjtDT1 in loquat fruit increased the malic acid content. Putatively, the inhibition of EjtDT1 transcription by EjAGL18, hinders the transport of malic acid from cytoplasm to vacuole and its concentration wihin this cell substructure.

Study is properly planned and performed. Obtained results support conclusions.  Results are novel,  original and could be interesting to researchers in the field. Following comments should be addressed before the publication.

1. Figure 3

Check out if the EjAGL18 should not be EjAGL181-263. If necessary correct appropriately Figure 3, description to Fig. 3 and the text of section 2.1.

2. Fig 4 and Fig 6

Add information in text, description to Fig 4 and 6,  as well as materials and methods if TRV-EjAGL18 #1,2,3 and OE-EjAGL18 #1,2,3 are just different transgenic lines obtained after transformation by  EjAGL181-263 or they are produced by transformation using EjAGL181-84, EjAGL1885-263, and EjAGL181-263 constructs.  

3. Section 4.2

Add following information to the text of section 4.2

I. How the purity and quality of RNA was assessed?

II. Provide approximate amount (ng) of RNA/cDNA for one qPCR sample

III. Provide details of qPCR reaction; temperature, duration, number of cycles

IV. Add in table S1 the length of amplicons (bp) for tested and reference gene used in qPCR.

Author Response

Dear reviewer,

Thank you very much for taking the time to review this manuscript. Your comments were important for us to improve our manuscript. We have studied your comments carefully and revised manuscript according to these comments. We hope our revised manuscript could meet your requirements. Please find the detailed responses below and the corresponding revisions highlighted using red font in the re-submitted files.

Comment 1: Authors identified a MADS-box transcription factor, EjAGL18, regulating the malic acid accumulation in loquat. Authors confirmed the nuclear localization of EjAGL18, however, the Y2H-based test showed the lack of trans-activation properties. The expression of EjAGL18 is inversely correlated with the malic acid concentration durig fruit ripening. Dual luciferase and Y1H assays showed that EjAGL18 binds to EjtDT1 promoter and repress its activity. Moreover, the transient overexpression of EjtDT1 in loquat fruit increased the malic acid content. Putatively, the inhibition of EjtDT1 transcription by EjAGL18, hinders the transport of malic acid from cytoplasm to vacuole and its concentration wihin this cell substructure.

Study is properly planned and performed. Obtained results support conclusions.  Results are novel, original and could be interesting to researchers in the field. Following comments should be addressed before the publication.

Response 1: Thanks very much for your positive affirmation and evaluation for our study. According to your comments, we have carefully revised the manuscript and the content of amendments was also highlighted using red font in our revised manuscript. The detailed point-by-point response to your comments is as follows.

Comment 2: Figure 3

Check out if the EjAGL18 should not be EjAGL181-263. If necessary correct appropriately Figure 3, description to Fig. 3 and the text of section 2.1.

Response 2: Thanks for your reminder. We have revised Figure 3 and text of section 2.1 accordingly.

Comment 3: Fig 4 and Fig 6

Add information in text, description to Fig 4 and 6, as well as materials and methods if TRV-EjAGL18 #1,2,3 and OE-EjAGL18 #1,2,3 are just different transgenic lines obtained after transformation by EjAGL181-263 or they are produced by transformation using EjAGL181-84, EjAGL1885-263, and EjAGL181-263 constructs.

Response 3: Thanks for your suggestion. 1#, 2# and 3# in this study represented three biological replicates of EjAGL18 transformed fruits. Each biological replicate consisted of six fruits. To avoid misunderstandings, we have added statements in the description to Figure 4 (L164) and in Materials and Methods section (L408).

Comment 4: Section 4.2

Add following information to the text of section 4.2

I.How the purity and quality of RNA was assessed?

II. Provide approximate amount (ng) of RNA/cDNA for one qPCR sample

III. Provide details of qPCR reaction; temperature, duration, number of cycles

IV. Add in table S1 the length of amplicons (bp) for tested and reference gene used in qPCR.

Response 4: Thanks for your advice. We have added corresponding content in section 4.2. In L359, we have added “RNA purity, quality and concentration were detected with 1 % RNase-free agarose gel electrophoresis and 2100 Bioanalyzer (Agilent Technologies, Santa Clara, CA, USA), respectively.” In L364 we have added “A reaction mixture of 10 µL included 5 µL SYBR mix, 0.2 µL gene-specific forward and reverse primers (10 µmol/L), 1 µL cDNA template (10 ng/µL), and 3.6 µL sterile water. The cycling procedure was as follows: 94 °C for 20 s; 94 °C for 10 s, 58 °C for 10 s, 72 °C for 10s, for 40 cycles.” According to your suggestion, we also added the length of amplicons (bp) of qPCR in Table S1.

Reviewer 3 Report

Comments and Suggestions for Authors

The manuscript by Chi et al. broadens the understanding of the MADS-box TF regulatory mechanisms on malic acid. In my opinion, the manuscript can be accepted after revision. Please see my comments to improve the manuscript.

-Line 31, “At present, commercial cultivation of loquat is widespread across numerous countries, such as China, Japan and Spain et al.’’. Is Spain a loquat producer?

-Line 105, RT-qPCR is the first appearance in the manuscript. It should be written in full name rather than an abbreviation.

-Line 110, “Figure 1. The expression pattern of EjAGL18” and in section 2.2, the EjAGL18 should be italic.

-Line 114, there is a lack of specific explanations of ‘CB_1’, ‘HYWH” and ‘WHZY’ abbreviations and illustrations of Figure 1.

-Line 175, the sequence of CarG box should be provided.

-To better understand how the downstream genes were screened, the transcriptional levels of EjNADP-MDH, EjNADP-ME, EjtDT1, EjtDT2 from Chi’s previous transcriptome data or RT-qPCR results should be provided in supplementary files.

- In Materials and Methods, there is a lack of conserved domain analysis methods.

Author Response

Dear reviewer,

Thank you very much for taking the time to review this manuscript. Your comments were important for us to improve our manuscript. We have studied your comments carefully and revised manuscript according to these comments. We hope our revised manuscript could meet your requirements. Please find the detailed responses below and the corresponding revisions highlighted using red font in the re-submitted files.

Comment 1: The manuscript by Chi et al. broadens the understanding of the MADS-box TF regulatory mechanisms on malic acid. In my opinion, the manuscript can be accepted after revision. Please see my comments to improve the manuscript.

Response 1: Thanks very much for your positive affirmation and evaluation for our study. According to your comments, we have carefully revised the manuscript and the content of amendments was also highlighted using red font in our revised manuscript. The detailed point-by-point response to your comments is as follows.

Comment 2: Line 31, “At present, commercial cultivation of loquat is widespread across numerous countries, such as China, Japan and Spain et al.’’. Is Spain a loquat producer?

Response 2: According to the record in literature, loquat is now commercially produced in many countries including China, Japan, India, Spain, Italy, Turkey, South Africa, the USA, Brazil, and Australia. China and Japan are the major producing countries. Spain is a loquat producer but also a chief exporting country of loquat (Lin et al., 1998; Janick et al., 2015). Similarly, Jing et al. (2023), Zou et al. (2020) and Wang et al. (2021) also mentioned that loquat is also planted in Spain. The references are as follows:

Janick, J., Zhang, Z., Lin, S., 2015. Important world cultivars of loquat. Acta Horticulturae, 1092, 25–32.

Jing, D. L., Liu, X. Y., He, Q., Dang, J. B., Hu, R. Q., Xia, Y., Wu, D., Wang, S. M., Zhang, Y., Xia, Q. Q., Zhang, C., Yu, Y. H., Guo, Q. G., Liang, G. L.,2023. Genome assembly of wild loquat (Eriobotrya japonica) and resequencing provide new insights into the genomic evolution and fruit domestication in loquat. Horticulture Research, 10, uhac265.

Lin, S., Sharpe, R. H., Janick, J.,1998. Loquat: Botany and horticulture. Horticultural Reviews, 23, 233-276.

Wang, Y.S., Paterson, A.H., 2021. Loquat (Eriobotrya japonica (Thunb.) Lindl) population genomics suggests a two-staged domestication and identifies genes showing convergence/parallel selective sweeps with apple or peach. The Plant Journal, 106, 942-952.

Zou, S. C., Wu, J. C., Shahid, M. Q., He, Y. H., Lin, S. Q., Liu, Z. W., Yang, X. H.,2020. Identification of key taste components in loquat using widely targeted metabolomics. Food Chemistry, 323.

Comment 3: Line 105, RT-qPCR is the first appearance in the manuscript. It should be written in full name rather than an abbreviation.

Response 3: Thanks for your reminder. We have added the full name of RT-qPCR in L106.

Comment 4: Line 110, “Figure 1. The expression pattern of EjAGL18” and in section 2.2, the EjAGL18 should be italic.

Response 4: According to your comment, we have italicized EjAGL18 in Figure 1 title and in section 2.2 (L152).

Comment 5: Line 114, there is a lack of specific explanations of ‘CB_1’, ‘HYWH” and ‘WHZY’ abbreviations and illustrations of Figure 1.

Response 5: Thanks for reminding us of this problem. We have added the explanation of the ‘CB_1’, ‘HYWH” and ‘WHZY’ abbreviations and illustrations in Figure 1 (L112). The revision and additions are as follows. “Figure 1. The expression pattern of EjAGL18 in ‘Wuhe Zaoyu’ (A), ‘Changbai No.1’ (B) and ‘Huayu Wuhe No.1’ (C) loquat fruits of different development stages. ‘WHZY’, ‘HYWH’ and ‘CB_1’ represented triploid loquat cultivar ‘Wuhe Zaoyu’, ‘Huayu Wuhe No.1’ and diploid loquat cultivar ‘Changbai No.1’, respectively. G3-G5 represented three development stages of loquat fruit, namely 21, 23, and 26 weeks post-anthesis. The gene expression was shown as bar chart and the malic acid content was shown as line graph. The values are shown as the mean ± SE.”

Comment 6: Line 175, the sequence of CarG box should be provided.

Response 6: According to your suggestion, we provided CarG box sequence of CC(A/T)6GG or C(A/T)8G in L186.

Comment 7: To better understand how the downstream genes were screened, the transcriptional levels of EjNADP-MDH, EjNADP-ME, EjtDT1, EjtDT2 from Chi’s previous transcriptome data or RT-qPCR results should be provided in supplementary files.

Response 7: Thanks for your instructive advice. Accordingly, we have additionally performed the RT-qPCR of EjNADP-MDH, EjNADP-ME, EjtDT1, EjtDT2 in loquat fruits at different development stages to better explain their high correlations with malic acid. The added RT-qPCR results were shown in Figure S2.

Comment 8: In Materials and Methods, there is a lack of conserved domain analysis methods.

Response 8: Thanks for your kind reminder. We have added “The conserved domains of EjAGL18 was analyzed in NCBI-CDD (https://www.ncbi.nlm.nih.gov/Structure/cdd/wrpsb.cgi).” in L375.

Reviewer 4 Report

Comments and Suggestions for Authors

Dear authors,

The findings were derived and obtained using appropriate analytical methods. The obtained results are convincing and clearly presented. I have some suggestions and a few minor remarks:

Suggestions:

1. Other than only transiently overexpression of EjtDT1 (pCAMBIA2300-EjtDT1) analysis, the silencing of EjAGL18 (pTRV2-EjAGL18 vector) and overexpressing of EjAGL18 (pCAMBIA2300-EjAGL18 vector) should be conducted in combination with overexpression of EjtDT1 by co-injection pCAMBIA2300-EjtDT1 with pTRV2-EjAGL18 vector or co-injection pCAMBIA2300-EjtDT1 with pCAMBIA2300-EjAGL18 vector as co-expressing analyses. Therefore, the research findings will be more robust. For instance, co-injection pCAMBIA2300-EjtDT1 vector with pTRV2-EjAGL18 vector might increase malate accumulation in loquat fruits than that of only injected with pCAMBIA2300-EjtDT1 vector.

2. The RT-qPCR of Fig.6B, the transcript level of EjtDT1 in overexpressing fruits (OE-EjtDT1) showed higher than those of control fruits (p2300). It could be because of the fact that the overexpressing fruits were injected with the transformed Agrobacterium GV310 harboring the OE-EjtDT1 vector. Thus these overexpressing fruits could present more EjtDT1 gene than control p2300. This might make the overexpressing fruits show a higher level of EjtDT1 mRNA since the fruits were heterogeneous overexpressing the EjtDT1 gene. Therefore, an RT-qPCR of EjAGL18 should be included, since overexpression of EjtDT1 downregulated/suppressed EjAGL18. For that, the OE-EjtDT1-injected fruits might show a lower relative expression level of the EjAGL18 gene.

Minor remarks

- Please pay special attention to the genes, transcription factors (TFs) and proteins/enzymes mentioned in the study, using italics to differentiate genes from TFs and proteins. CHS, PeCHS,... etc. distinguish between the spellings for gene and protein.  For  gene,  it  pesent  in  italic. However, TF and protein should not be presented in italic. For example, “EjAGL18” (Line 146-149 and Fig.4’s legend) denotes for gene, therefore, it should be presented in italic. Please double-check the whole manuscript and revise it.

- Please provide GenBank accession number for the genes (EjAGL18, qEjActin, qEjAGL18, qEjtDT2, qEjNADP-MDH, qEjNADP-ME, qEjtDT1) listed in Table S1.

- Overall, the manuscript is readable and easy to understand except for a few word grammar errors and sentence structure. Please thorough review and revise them.

For instance:

+ Spain et al. [1] (Line 32)

+ Metabolic enzymes a play crucial role in these processes. (L48-49)

+ Errors: explication/notation “Reportor” in Fig.5F. It should be “Reporter”.

+ Revise “BD-EjAGL181-84” (Line 129) to “BD-EjAGL181-84” and “BD-EjAGL1885-263” (L130) to “BD-EjAGL1885-263

- Italicize: “Agrobacterium” (L391)

 I have yellow-marked some of the above minor remarks on the manuscript. Please use it for easy tracking and revision.

Best regards,

Author Response

Dear reviewer,

Thank you very much for taking the time to review this manuscript. Your comments were important for us to improve our manuscript. We have studied your comments carefully and revised manuscript according to these comments. We hope our revised manuscript could meet your requirements. Please find the detailed responses below and the corresponding revisions highlighted using red font in the re-submitted files.

Comment 1: The findings were derived and obtained using appropriate analytical methods. The obtained results are convincing and clearly presented. I have some suggestions and a few minor remarks:

Response 1: Thanks very much for your positive affirmation and evaluation for our study. According to your comments, we have carefully revised the manuscript and the content of amendments was also highlighted using red font in our revised manuscript. The detailed point-by-point response to your comments is as follows.

Comment 2: Other than only transiently overexpression of EjtDT1 (pCAMBIA2300-EjtDT1) analysis, the silencing of EjAGL18 (pTRV2-EjAGL18 vector) and overexpressing of EjAGL18 (pCAMBIA2300-EjAGL18 vector) should be conducted in combination with overexpression of EjtDT1 by co-injection pCAMBIA2300-EjtDT1 with pTRV2-EjAGL18 vector or co-injection pCAMBIA2300-EjtDT1 with pCAMBIA2300-EjAGL18 vector as co-expressing analyses. Therefore, the research findings will be more robust. For instance, co-injection pCAMBIA2300-EjtDT1 vector with pTRV2-EjAGL18 vector might increase malate accumulation in loquat fruits than that of only injected with pCAMBIA2300-EjtDT1 vector.

Response 2: Thanks for your instructive advice. According to your advice, we additionally performed the transient co-transformation experiments of pCAMBIA2300-EjtDT1 vector and pTRV2-EjAGL18 vector in loquat fruits. The results showed that overexpressing EjtDT1 and silencing EjAGL18 at the same time significantly increased the EjtDT1 transcriptional level and malic acid content compared to overexpressing EjtDT1 alone. This result provided further evidence to support our study.

Accordingly, the supplementary co-transformation results were updated in Figure 6 and content in section 2.4. The added content is as follows (L224). “To further clarify that the EjAGL18 affected the accumulation of malic acid by direct regulating the expression of EjtDT1, the pCAMBIA2300-EjtDT1 recombinant vector and pTRV2-EjAGL18 recombinant vector were transiently co-transformed into loquat fruits, with the empty vectors pCAMBIA2300 and pTRV2 as controls (Figure 6A). The results showed that the overexpression of EjtDT1 and simultaneous silencing of EjAGL18 significantly increased the EjtDT1 transcriptional level and malic acid content compared to overexpression of EjtDT1 alone (Figure 6B-D). These results further confirmed that EjAGL18 could bind to the promoter of EjtDT1, repressing its expression and thereby affecting the accumulation of malic acid in loquat fruits.” In discussion (L327), we have added “When EjAGL18 was silenced in EjtDT1-overexpressing fruits, the content of malic acid was significantly higher than in fruits that overexpressing EjtDT1 alone.”

Comment 3: The RT-qPCR of Fig.6B, the transcript level of EjtDT1 in overexpressing fruits (OE-EjtDT1) showed higher than those of control fruits (p2300). It could be because of the fact that the overexpressing fruits were injected with the transformed Agrobacterium GV310 harboring the OE-EjtDT1 vector. Thus these overexpressing fruits could present more EjtDT1 gene than control p2300. This might make the overexpressing fruits show a higher level of EjtDT1 mRNA since the fruits were heterogeneous overexpressing the EjtDT1 gene. Therefore, an RT-qPCR of EjAGL18 should be included, since overexpression of EjtDT1 downregulated/suppressed EjAGL18. For that, the OE-EjtDT1-injected fruits might show a lower relative expression level of the EjAGL18 gene.

Response 3: Thanks for your suggestion. We additional tested the transcriptional level of EjAGL18 in EjtDT1 overexpressing fruits. As shown in revised Figure 6C, the expression level of EjAGL18 presented no significant difference in overexpression fruits of EjtDT1 compared to control. It suggested that high mRNA level of EjtDT1 did not suppress or activate EjAGL18. We speculated that this was probably because the transcription of EjAGL18 was independent to EjtDT1 or regulated by other factors, which needs to be further investigated in the future.

Comment 4: Please pay special attention to the genes, transcription factors (TFs) and proteins/enzymes mentioned in the study, using italics to differentiate genes from TFs and proteins. CHS, PeCHS,... etc. distinguish between the spellings for gene and protein. For gene, it pesent in italic. However, TF and protein should not be presented in italic. For example, “EjAGL18” (Line 146-149 and Fig.4’s legend) denotes for gene, therefore, it should be presented in italic. Please double-check the whole manuscript and revise it.

Response 4: Thanks for your reminder. We have double-checked the entire manuscript and revised the font accordingly.

Comment 5: Please provide GenBank accession number for the genes (EjAGL18, qEjActin, qEjAGL18, qEjtDT2, qEjNADP-MDH, qEjNADP-ME, qEjtDT1) listed in Table S1.

Response 5: Thanks for your advice. We have listed the gene IDs of all genes involved in this study in Table S1. Notably, the gene IDs of EjAGL18, EjtDT1/2, EjNADP-MDH, EjNADP-ME were from our published genome data (Jing et al., 2023) rather than GeneBank accession number. Our genome data could be found in National Genomics Data Center repository (Accession No. GWHBOTF00000000).

Jing, D.L.; Liu, X.Y.; He, Q.; Dang, J.B.; Hu, R.Q.; Xia, Y.; Wu, D.; Wang, S.M.; Zhang, Y.; Xia, Q.Q.; et al. Genome assembly of wild loquat (Eriobotrya japonica) and resequencing provide new insights into the genomic evolution and fruit domestication in loquat. Hortic. Res. 2023, 10, uhac265. https://doi.org/10.1093/hr/uhac265.

Comment 6: Overall, the manuscript is readable and easy to understand except for a few word grammar errors and sentence structure. Please thorough review and revise them.

For instance:

+ Spain et al. [1] (Line 32)

+ Metabolic enzymes a play crucial role in these processes. (L48-49)

+ Errors: explication/notation “Reportor” in Fig.5F. It should be “Reporter”.

+ Revise “BD-EjAGL181-84” (Line 129) to “BD-EjAGL181-84” and “BD-EjAGL1885-263” (L130) to “BD-EjAGL1885-263”

- Italicize: “Agrobacterium” (L391)

 I have yellow-marked some of the above minor remarks on the manuscript. Please use it for easy tracking and revision.

Response 6: Thank you so much for your careful review of our manuscript. We have corrected the mistakes that you’ve marked in the manuscript. Meanwhile, we have checked the manuscript thoroughly to ensure that there are no more grammatical errors.

Round 2

Reviewer 1 Report

Comments and Suggestions for Authors

Thanks for authors works, the review had been well revised, most of my comments were well addressed in revision, and I have no new comments about it. Good luck.

Reviewer 3 Report

Comments and Suggestions for Authors

The authors addressed all the comments and improved the manuscript. In my opinion, it can be accepted in its present form.